Expression of IER3 in hepatocellular carcinoma: clinicopathology, prognosis, and potential regulatory pathways

He Fei-Yan 1
Chen Gang 1
He Rong-quan 2
Huang Zhi-Guang 1
Li Jian-Di 1
Wu Wei-Zi 3
Chen Ji-Tian 3
Tang Yu-Lu 1
Li Dong-Ming 1
Pan Shang-Ling 4
Feng Zhen-Bo 1
Dang Yi-wu dangyiwu@126.com 1
1 Department of Pathology, First Affiliated Hospital of Guangxi Medical University , Nanning , Guangxi Zhuang Autonomous Region , P.R. China
2 Department of Medical Oncology, First Affiliated Hospital of Guangxi Medical University , Nanning , Guangxi Zhuang Autonomous Region , P.R. China
3 Department of Pathology, People’s Hospital of Ling Shan , Ling Shan , Guangxi Zhuang Autonomous Region , P.R. China
4 Department of Pathophysiology, School of Pre-clinical Medicine, Guangxi Medical University , Nanning , Guangxi Zhuang Autonomous Region , P.R. China
Sotelo-Mundo Rogerio
Electronic publication date: 2022 Mar 10
Publication date: 2022
Volume: 10
Electronic Location ID: e12944
Received 2021 Aug 27; Accepted 2022 Jan 24
Copyright: ©2022 He et al.
Copyright year: 2022
Copyright holder: He et al.
License: This is an open access article distributed under the terms of the Creative Commons Attribution License, which permits unrestricted use, distribution, reproduction and adaptation in any medium and for any purpose provided that it is properly attributed. For attribution, the original author(s), title, publication source (PeerJ) and either DOI or URL of the article must be cited.
License URL: https://creativecommons.org/licenses/by/4.0/

Keywords: Immediate early response 3 (IER3), Hepatocellular carcinoma (HCC), Immunohistochemistry, Gene chip, RNA-sequencing, Hazard ratios (HRs)

Funding: National Natural Science Foundation of China NSFC 81860717 Guangxi Medical High-level Key Talents Training “139” Program (2020) Guangxi Medical University Training Program for Distinguished Young Scholars (2017) Guangxi Zhuang Autonomous Region Health Committee Self-financed Scientific Research Project Z20190523 This research was supported by National Natural Science Foundation of China (NSFC 81860717), Guangxi Medical High-level Key Talents Training “139” Program (2020), Guangxi Medical University Training Program for Distinguished Young Scholars (2017), Guangxi Zhuang Autonomous Region Health Committee Self-financed Scientific Research Project (Z20190523). The funders had no role in study design, data collection and analysis, decision to publish, or preparation of the manuscript.

==============================
Background

Immediate early response 3 (IER3) is correlated to the prognosis of several cancers, but the precise mechanisms underlying the regulation by IER3 of the occurrence and development of hepatocellular carcinoma (HCC) remain unknown.

Methods

The expression level of IER3 was examined by using in-house immunohistochemistry (IHC), public gene chip, and public RNA-sequencing (RNA-seq). The standardized mean difference (SMD) was calculated to compare the expression levels of IER3 between HCC patients and controls. The summary receiver operating characteristics (sROC) was plotted to comprehensively understand the discriminatory capability of IER3 between HCC and non-HCC group. The Kaplan–Meier curves and the combined hazard ratios (HRs) were used to determine the prognostic value of IER3 in HCC. Moreover, differentially expressed genes (DEGs) and co-expression genes (CEGs) were used to explored the molecular mechanisms of IER3 underlying HCC. hTFtarget was used to predict the transcription factors (TFs) of IER3. The binding site of TFs and the IER3 promoter region was forecasted using the JASPAR website. The relevant ChIP-seq data were used to determine whether TF peaks were present in the IER3 transcription initiation.

Results

A significantly increased expression of IER3 protein was found in HCC tissue relative to non-HCC tissue as detected by IHC (p < 0.001). Compared to 1,263 cases of non-HCC tissues, IER3 in 1483 cases of HCC tissues was upregulated (SMD = 0.42, 95% confidence interval [CI] [0.09–0.76]). The sROC showed that IER3 had a certain ability at differentiating HCC tissues (area under the curve (AUC) = 0.65, 95% CI [0.61–0.69]). Comprehensive analysis of the effect of IER3 on the prognosis of patients with HCC demonstrated that higher IER3 expression was associated with poor prognosis in HCC (HRs = 1.30, 95% CI [1.03–1.64]). Pathway enrichment analysis revealed that IER3-related genes were mostly enriched in the PI3K-Akt signaling pathway, cancer-related signaling pathways, the p53 signaling pathway, and other signaling pathways. Regulatory factor X5 (RFX5) was identified as a possible regulator of IER3-related TF.

Conclusion

IER3 may be a potential prognostic marker for HCC. The molecular mechanisms of IER3 in HCC warrant further study.

Introduction

Primary hepatic carcinoma (PHC) is one of the most common malignant cancers, ranking third in cancer-related deaths worldwide (Chang et al., 2020; Sung et al., 2021). Hepatocellular carcinoma (HCC) is the most common pathological type of PHC, accounting for 75% to 95% of all PHC cases (Sung et al., 2021). The leading pathogenies for HCC are hepatitis B virus (HBV) or hepatitis C virus (HCV), ingesting food contaminated with aflatoxin, heavy drinking, overweight, type 2 diabetes, and smoking (Anwanwan et al., 2020; Dakurah et al., 2021; Elpek, 2021; Jia et al., 2020; McGlynn, Petrick & El-Serag, 2021; Wang et al., 2021). At present, surgery is the main measure for the treatment of HCC. The outcome of early surgical resection is better than that of late surgical resection, and the postoperative survival time of patients is longer. However, for cases of middle-and late-stage, large tumors, and multiple tumors, the radical resection rate is low and the prognosis is poor. Even in cases in which the primary tumor can be completely surgically removed treated post-operatively with radiotherapy and chemotherapy, the recurrence rate of HCC remains extremely high (Liu et al., 2020; Luo et al., 2020; Trevisan França de Lima et al. 2020; Yang & Heimbach, 2020). In order to improve the efficiency of diagnosis and the effectiveness of prevention and treatment of HCC, the etiological underpinnings and pathogenic mechanisms of HCC must be understood in greater detail. Hence, there is an urgent need to identify novel targets of therapy to improve prognosis in HCC.

Immediate early response 3 (IER3), also known as IEX1, belongs to the immediate early response gene family. The constituents of this gene family can be transcribed and activated within a few minutes, and their peak expression can be reached within 15–20 min following stimulation, without the need for ab initio protein synthesis and expression. Many members of this gene family are transcription factors that can rapidly activate gene transcription, while others are essential for cells to respond to stressors promptly (Arlt & Schäfer, 2011; Wu et al., 2013). The growing body of evidence suggests that IER3 is correlated to the prognosis of various cancers (Wu et al., 2013), including bladder cancer (Ye et al., 2018), ovarian cancer (Han et al., 2011), and pancreatic cancer (Sasada et al., 2008).

With respect to a study of IER3 in HCC, IER3 knockdown inhibited the viability, growth, and migration of HCC cells (Emma et al., 2016). In the study of drug resistance of HCC, E26 transformation-specific (ETS) variant transcription factor 4 (ETV4) may regulate cell survival and proliferation through IER3 under the stimulation of sorafenib or cisplatin (Chen et al., 2019). The glycolysis-related gene pairs (GRGPs) can predict the prognosis of HCC, and galactokinase 1 (GALK1) and IER3 can be effective prognostic factors for HCC (Zhou et al., 2020). Studies have revealed that IER3 knockdown may affect the progression of HCC. The expression of IER3, a target gene of triacetylacetone proline (TTP), was decreased in HCC cells treated by MAPKAP2 (MK2) inhibitors, suggesting that IER3 may be one of the therapeutic targets for HCC. The current literature focuses predominantly on the relationship between IER3 in HCC and the clinical pathological parameters. The expression of IER3 in HCC was related to tumor progression; however, the precise mechanisms underlying the regulation by IER3 of the occurrence and development of HCC remain unknown. IER3 is not a transcription factor (TF), because it lacks a DNA binding domain. However, IER3 can sometimes be used as a co-activator or co-inhibitor, and it plays a key role in regulating apoptosis, proliferation, differentiation, and metabolism.

The objective of this study is to probe the clinical value of IER3 in HCC and its potential regulatory network. The expression level of IER3 was verified by using immunohistochemical staining to determine the prognostic value of IER3 in HCC, thereby paving the way for future clinical application. We analyzed the gene chip and RNA sequencing (RNA-seq) data sets, focusing on the expression of IER3 messenger RNA (mRNA) in HCC. Moreover, we also explored the molecular mechanisms of IER3 underlying HCC and predicted the TFs that could bind to the IER3 promoter region, so as to ultimately broaden our understanding of IER3 in the incidence and progression of HCC.

Materials & Methods

HCC sample collection and IHC

A total of 94 paraffin masses of HCC and adjacent tissues and 33 cases of normal liver tissues were collected. The expression of the IER3 protein was detected by immunohistochemistry (IHC) in 94 cases of tumor tissues and in 127 cases of non-tumor tissues. The following information was supplied relating to ethical approvals: The First Affiliated Hospital of Guangxi Medical University approved this study (2018 KY-E-102). All study participants agreed to the study and received written informed consent from study participants. The tissue samples were fixed by formalin, dewaxed with an environmental dewaxing agent, and then repaired with ethylenediaminetetraacetic acid (EDTA) repair solution for antigen repair. Endogenous peroxidase was inactivated with 3% H2O2, and tissue samples were incubated with an anti-IER3 antibody in a wet box for 90 min at 37 °C, and incubated at room temperature with Supervision™ mouse/rabbit universal secondary antibody detection system for 20 min. The tissue sections were stained with a 3,3-diaminobenzidine (DAB) chromogenic kit, and then stained with hematoxylin. Finally, images were captured via microscopy. The staining results were scored by two pathologists at our institution using a double-blind procedure. The staining intensity was categorized as either absent (0), weak (1), medium (2), or strong (3). The proportions of positive cells were 0% (0), 1–25% (1), 26–50% (2), 51–75% (3), and 76–100% (4), respectively. The IHC score was the product of the staining intensity and the proportions of positive cells (Pang et al., 2020b). The clinicopathological parameters of each patient were recorded and included age, gender, pathological grade, alcohol history, and alpha-fetoprotein (AFP). The relationship between IER3 protein expression and clinicopathological characteristics was determined using independent sample’s t-test and Pearson correlation test in SPSS v.25.0. To determine the prognostic significance of IER3, the survival rates of patients with high and low IER3 expression were compared using the Kaplan–Meier curve. The median expression level was used to distinguish the higher from the lower IER3 expression group. A p value < 0.05 was deemed statistically significant.

Collection of data from public databases

The microarrays and RNA-seq data of HCC were searched in the Gene Expression Omnibus (GEO) database, the Oncomine database, and in the scientific literature. The search keywords were: ((hepatocellular OR liver) AND (cancer OR tumor OR carcinoma)) OR ((HCC) OR (Hepatocellular Carcinoma)). The inclusion criteria were: (1) the collected sample tissue was HCC tissue; (2) the sample tissue source was Homo sapiens; (3) the number of tumors and non-tumor samples were ≥3; and (4) the expression levels of IER3 could be obtained. The exclusion criteria were datasets that do not meet the above requirements. We merged GEO Series (GSE) data sets according to various GEO Platform (GPL), and used the limma voom function in R v.3.6.1 to eliminate batch effects between studies. In addition, a log2 (x+1) transformation was performed when matrices were not normalized

RNA-seq data collection from TCGA database and GTEx

The RNA-seq data set of HCC was downloaded and extracted from The Cancer Genome Atlas Program (TCGA) database. RNA-seq data from normal liver tissues were downloaded in the Genotype-Tissue Expression (GTEx) Project. The two datasets were merged, the batch effect was eliminated by the limma voom function in R v.3.6.1, and the resultant dataset was normalized by log2 (x+1). We also extracted the clinicopathological parameters of HCC patients from the TCGA database for follow-up analysis, including age, gender, pathological grade, pathological stage, alcohol history, and survival data, among others.

Comprehensive analysis of HCC data sets

The expression level data of IER3 were extracted in microarrays, RNA-seq datasets, and in-house IHC. The datasets were divided into the cancer tissue group and non-cancer tissue group. The standardized mean difference (SMD) was calculated with STATA v.14.0 software to compare the expression levels of IER3 between HCC patients and controls. The heterogeneity between the included studies was expressed by I2, and I2 ≥ 50% with a p value < 0.05 was considered statistically significant, in which case a random effect model was used. The publication bias was examined to evaluate the stability of SMD, and sensitive forest maps were drawn to explore the potential sources of heterogeneity. True positive rate (tp), false positive rate (fp), true negative rate (fn), and false negative rate (tn) for each data set were calculated according to the Jordan Index (Yoden Index), and STATA v.14.0 was used to determine the summary receiver operating characteristics (sROC) of all data sets. The total area under the curve (AUC) was calculated. and the sensitivity and specificity maps were forecasted (Li et al., 2020).

Prognostic value of IER3 in HCC

To determine the prognostic value of IER3 in HCC, the research gathered the clinicopathological features of HCC and determined differences in IER3 expression between the two groups using the independent samples t-test. The correlations between IER3 expression and various clinicopathological parameters were calculated with the Pearson correlation test. The survival rates of patients with high or low IER3 expression were compared using the Kaplan–Meier curve. The logarithmic rank test was used to determine whether the difference in prognosis was fulfilled. Then, the clinicopathological features of the TCGA data set were used to compare the correlation between the expression of IER3 and clinical features, and the Kaplan–Meier (K-M) curve was used to compare the survival rate between the high and low IER3 expression groups. To ensure that our results were consistent with those of other studies, we searched for the expression and overall survival (OS) of IER3 in HCC tissues from data on The Human Protein Atlas (https://www.proteinatlas.org/). The present study also explored the effect of IER3 expression on OS rates of HCC patients using the GEPIA (http://gepia.cancer-pku.cn/), the Kaplan–Meier Plotter (https://kmplot.com/analysis/index.php?p=service), SurvExpress (Aguirre-Gamboa et al., 2013), and the Biomedical Informatics Institute (An et al., 2020). The hazard ratios (HRs) were merged using STATA v.14.0 software (Ioannou, 2021). In GEPIA and Kaplan–Meier Plotter, we searched for the disease-free survival (DFS), progression free survival (PFS), and relapse free survival (RFS) of IER3 in HCC tissues.

Genetic variations of IER3

The main mutation types of IER3 in HCC tissues were obtained in cBioPortal for Cancer Genomics (https://www.cbioportal.org/). Information about the relationship between IER3 gene expression changes and the OS and disease-free survival (DFS) of patients with HCC were also analyzed.

Identification of DEGs and IER3 CEGs in HCC

According to the expression matrix of HCC tissue and non-cancer tissue, the limma voom function in R v3.6.1 software was used to screen for differentially expressed genes (DEGs). The screening condition for DEGs was that the log2 (fold change) > 1 and p < 0.05. The screening condition for co-expression genes (CEGs) was that the Pearson correlation coefficient ≥0.3 and p < 0.05. For all data, including TCGA and GSE datasets, we calculated CEGs and DEGs.

Potential mechanisms of IER3 underlying HCC

We intersected the upregulated DEGs and CEGs, which were used for functional annotation in HCC. CEGs and DEGs appeared more than three times in the mRNA datasets. The crossover genes were deposited into the Database for Annotation, Visualization, and Integrated Discovery (DAVID) and analyzed by Gene Ontology (GO) and Kyoto Encyclopedia of Genes and Genomes (KEGG). The Sangerbox tool (http://sangerbox.com/tool) was used to visualize the 10 most important biological process (BP), cellular components (CC), molecular function (MF), and KEGG projects of gene enrichment (Tang et al., 2020). Then we used the data in TCGA to sort each gene according to logFC value and carried out Gene Set EnrichmentAnalysis (GSEA) analysis in R software.

Potential IER3 transcription factors

The hTFtarget (http://bioinfo.life.hust.edu.cn/hTFtarget) collected the most comprehensive data of all human TF targets, which we used to predict the TFs of IER3. The positively related genes of TFs and IER3 were intersected to identify the TFs that could be associated with IER3. The binding site of TFs and the IER3 promoter region were forecasted using the JASPAR website (http://jaspar.genereg.net/). The Cistrome DB website (http://cistrome.org/db/#/) and The Integrative Genomics Viewer (IGV v.2.10.0) were used to determine whether TF peaks were presented in the IER3 transcription initiation. Subsequently, we also used mRNA datasets from public databases to explore the expression of TFs in HCC.

Statistical analysis

In this study, the software SPSS v.25.0, STATA v.14.0, R v.3.6.1, and GraphPad v.8.0.2 were used for statistical analysis and chart drawing. SPSS v.25.0 was used to compare the differences between the two groups using the independent samples t-test. A p < 0.05 was regarded as statistically significant.

Results

IER3 is highly expressed in HCC

As shown in Fig. 1, a general and significantly increased expression of IER3 was found in HCC tissue relative to non-HCC tissue (p < 0.001). Table 1 confirmed that the higher expression of IER3 was positively correlated with age (≥60 years old). However, the expression of IER3 was not related to other clinicopathological parameters. More importantly, according to the Kaplan–Meier curve based on IHC, the high level of IER3 was positively correlated with poor survival rate in HCC (Fig. 2A), but this result was not statistically significant (p > 0.05).

Figure 1 IER3 protein expression levels in HCC.

(A) Normal liver tissue sample #1 for IER3, IHC (×100, ×200, ×400), IHC score = 0. (B) Normal liver tissue sample #2 for IER3, IHC (×100, ×200, ×400), IHC score =2. (C) HCC tissue sample #1 for IER3, IHC (×100, ×200, ×400), IHC score =12. (D) HCC tissue sample #2 for IER3, IHC (×100, ×200, ×400), IHC score = 12. (E) IHC score analysis of IER3 in HCC samples and normal tissue samples. HCC, hepatocellular carcinoma; IER3, immediate early response 3; IHC, immunohistochemistry.

Table 1 The relationship between IER3 expression and clinicopathological arguments of HCC based on in-house IHC.

Clinicopathologic arguments	Number	IER3 expression	R	p-value	
Tissue	HCC	94	10.64 ± 2.08	0.725	<0.001	
Non-HCC	127	6.50 ± 2.73			
Age	<60	171	8.03 ± 3.24	0.843	<0.001	
≥60	50	9.04 ± 3.00			
Gender	Female	64	7.61 ± 3.67	–0.053	0.677	
Male	157	8.52 ± 2.97			
Pathologic grade	I–II	63	10.73 ± 1.88	–0.164	0.353	
III–IV	24	10.67 ± 2.55		
Alcohol history	No	54	9.37 ± 3.10	0.211	0.128	
Yes	53	10.64 ± 2.21		
Alpha-fetoprotein (AFP)	<20 ng/mL	44	8.86 ± 3.30	–0.076	0.021	
>20 ng/mL	62	10.77 ± 2.12		
Notes.

HCC hepatocellular carcinoma

IER3 immediate early response 3

IHC immunohistochemistry

R Pearson correlation coefficient

Figure 2 Prognostic values of IER3 in assessed by HCC.

(A) In-house IHC. (B) TCGA database. (C) Overall survival HR was calculated based on five data sets. HCC, hepatocellular carcinoma; IHC, immunohistochemistry; TCGA, The Cancer Genome Atlas; HR, hazard ratio.

Upregulation of IER3 mRNA expression in HCC

A total of 34 GSE data sets were merged into 16 platform matrices, and the combined data of TCGA, GTEx, and in-house IHC data included 1483 cases of HCC tissues and 1263 cases of non-cancer tissues. The detailed information of the included matrices is shown in Table 2. Figure 3A shows that, compared to non-HCC tissues, IER3 mRNA in HCC tissues was upregulated (SMD = 0.42, 95% CI [0.09–0.76], p < 0.001). Due to significant heterogeneity (I2 = 91.6%, p < 0.001), the random effects model was used. The Egger’s test (Fig. 3B) found no significant publication bias (p = 0.506). Sensitivity analysis (Fig. 3C) revealed that the studies included did not explain the source of the heterogeneity. The sROC (Fig. 4A) showed that IER3 mRNA had a relatively weak ability at differentiating HCC tissues (AUC = 0.65, 95% CI [0.61–0.69]), and the corresponding sensitivity and specificity (Fig. 4B) were 0.60 (95% CI [0.48–0.71]) and 0.64 (95% CI [0.43–0.81]), respectively. The positive and negative diagnostic likelihood ratio (DLR) (Fig. 4C) was 1.67 (1.02–2.74) and 0.62 (0.45–0.85), respectively.

Table 2 Basic characteristics of IER3 expression.

GEO platform	Data source	HCC tissue	Non-HCC tissue	p-value	
		Number	Mean	SD	Number	Mean	SD		
GPL5175	GSE12941, GSE84005	48	3.04	0.13	48	3.05	0.12	0.724	
GPL6244	GSE45050, GSE64041	68	3.99	0.25	73	3.90	0.25	0.027	
GPL6480	GSE117361, GSE54236	160	3.49	0.14	5	3.46	0.11	0.644	
GPL11154	GSE63863, GSE65485, GSE73708, GSE81550, GSE87592, GSE174608, GSE114564, GSE148355, GSE69164	195	2.99	0.89	150	2.94	0.54	0.556	
GPL16791	GSE104310, GSE63018, GSE77509, GSE94660, GSE97214, GSE112221	95	6.26	0.89	77	6.05	0.88	0.124	
GPL21047	GSE101728, GSE98269	10	3.68	0.12	10	3.67	0.11	0.921	
GPL5474	GSE10143	80	13.76	0.73	307	12.41	0.86	<0.001	
GPL3921	GSE14520	229	3.03	0.22	216	3.04	0.24	0.725	
GPL10999	GSE33294	3	4.88	1.64	3	3.74	0.54	0.317	
GPL4133	GSE46408	6	11.50	1.02	6	10.28	1.09	0.073	
GPL13369	GSE46444	88	10.34	1.62	48	10.36	1.61	0.939	
GPL16699	GSE57555	5	0.03	0.18	5	–0.07	0.05	0.273	
GPL96	GSE60502	18	10.46	1.35	18	10.17	0.77	0.427	
GPL20301	GSE125469	3	6.90	0.69	3	5.98	0.55	0.143	
GPL18573	GSE128274	4	4.24	0.93	4	4.21	0.99	0.964	
GPL23126	GSE166163	3	6.30	0.56	3	6.09	1.73	0.845	
TGCA-GTEx	–	374	5.59	1.49	160	5.58	0.95	0.945	
In-house IHC	–	94	10.64	2.08	127	6.50	2.73	<0.001	
Notes.

HCC hepatocellular carcinoma

IER3 immediate early response 3

IHC immunohistochemistry

TCGA The Cancer Genome Atlas

GTEx Genotype-Tissue Expression

SD standard deviation

Figure 3 Comprehensive expression of IER3 in HCC calculated with SMD.

(A) SMD forest map. (B) Egger’s test. (C) sensitivity analysis. HCC, hepatocellular carcinoma; SMD, standardized mean difference.

Figure 4 The comprehensive performance of IER3 to distinguish HCC from non-HCC.

(A) sROC. (B) Sensitivity and specific forest map. (C) DLR Positive and DLR Negative forest map. HCC, hepatocellular carcinoma; IER3, immediate early response 3; sROC, summary receiver operating characteristics; DLR, diagnostic likelihood ratio.

High IER3 expression in HCC is associated with poor HCC prognosis

The higher level of IER3 in RNA-seq and clinicopathological parameters were positively correlated with pathologic grade (G3-G4), pathologic stage (III–IV), new tumor event after initial treatment (yes), and alcohol history (Table 3). The K-M curve revealed that the high expression of IER3 predicted poor HCC prognosis (p = 0.019; Fig. 2B). After merging the overall survival HRs of five data sets (GSE10143, GSE76427, GSE27150, TCGA, and in-house IHC), the total HR was 1.30 (95% CI [1.03–1.64]), indicating that high IER3 expression could be used as a pathogeny factor for overall survival of HCC (Fig. 2C). Univariate and multivariate Cox-regression results revealed that new tumor event after initial treatment was connected with OS in the univariate analysis (p <0.001). And new tumor event after initial treatment might be an independent prognostic factor (Table 4). The DFS curve suggested that high expression of IER3 may be associated with increased disease recurrence rate (p = 0.048, Fig. 5A; p = 0.007, Fig. 5B). The PFS curve suggested that the expression of IER3 may be associated with the progression-free survival of HCC, but the results were not statistically significant (p = 0.086, Fig. 5C). The RFS curve suggested that the expression of IER3 may be associated with the relapse free survival of HCC, but the results were not statistically significant (p = 0.078, Fig. 5D).

Table 3 The relationship between IER3 expression and clinicopathological arguments of HCC based on the TCGA database.

Clinicopathologic arguments	Number	IER3 expression	R	p- value	
Age	≥60	232	10.79 ± 0.91	0.006	0.938	
<60	191	8.16 ± 0.94			
Gender	Male	280	9.53 ± 1.61	0.075	0.389	
Female	143	9.73 ± 1.61			
Pathologic grade	G1–G2	265	9.47 ± 1.61	0.18	0.027	
G3–G4	150	9.82 ± 1.60			
Pathologic stage	I–II	288	9.56 ± 1.61	0.677	<0.001	
III–IV	103	9.66 ± 1.65			
New tumour event after initial treatment	No	196	9.49 ± 1.50	0.995	<0.001	
Yes	198	9.65 ± 1.68			
Alcohol history	No	275	9.58 ± 1.67	0.265	0.003	
Yes	126	9.60 ± 1.59			
Notes.

HCC hepatocellular carcinoma

IER3 immediate early response 3

TCGA The Cancer Genome Atlas

R Pearson correlation coefficient

Table 4 Univariate and multivariate Cox-regression results of factors related to overall survival based on TCGA.

Univariate analysis				Multivariate analysis		
Variables		Hazard ratio (95% CI)	p value	Hazard ratio (95% CI)	p value	
Age	<50	1.000 (reference)		1.000 (reference)		
	50–59	1.190(0.815–1.737)	0.369	1.190(0.801–1.769)	0.389	
	60–69	1.172(0.812–1.693)	0.397	1.274(0.871–1.862)	0.212	
	70–79	0.682(0.4375–1.063)	0.091	0.660(0.411–1.059)	0.085	
	≥80	1.098(0.393–3.067)	0.858	1.420(0.494–4.084)	0.516	
Gender	Female	1.000 (reference)	–	1.000 (reference)	–	
	Male	1.362(1.018–1.822)	0.038	1.274(0.925–1.756)	0.139	
Pathologic grade	G1	1.000 (reference)		1.000 (reference)		
	G2	0.989(0.664–1.474)	0.958	1.207(0.791–1.842)	0.384	
	G3	0.949(0.623–1.444)	0.806	1.149(0.731–1.806)	0.549	
	G4	1.384(0.632–3.031)	0.806	1.331(0.576-3.078)	0.503	
Pathologic stage	Stage I	1.000 (reference)	–	1.000 (reference)	–	
	Stage II	1.109(0.809–1.519)	0.520	1.203(0.869–1.667)	0.266	
	Stage III	0.790(0.559–1.117)	0.183	0.904(0.633–1.291)	0.580	
	Stage IV	0.886(0.124–6.357)	0.904	1.333(0.180–9.894)	0.779	
New tumor event after initial treatment	No	1.000 (reference)	–	1.000 (reference)	–	
	Yes	0.548(0.420–0.715)	<0.001	0.522(0.396–0.688)	<0.001	
Alcohol history	No	1.000 (reference)	–	1.000 (reference)	–	
	Yes	1.169(0.886–1.542)	0.268	1.140(0.842–1.544)	0.397	

Figure 5 The Kaplan–Meier curves of HCC.

(A) DFS curve in GEPIA. (B) DFS curve in Kaplan–Meier Plotter. (C) PFS curve in Kaplan–Meier Plotter. (D) RFS curve in Kaplan–Meier Plotter. HCC, hepatocellular carcinoma. DFS, disease-free survival; PFS, progression free survival; RFS, relapse free survival.

Genetic changes and mutation types of IER3 in HCC

In the OncoPrint of IER3 alterations, 17 genetic changes were observed in HCC samples, of which 16 were amplification and 1 was a missense mutation (accounting for 1.9%; Fig. 6A). With respect to the relationship between IER3 alterations and total OS of HCC patients (Fig. 6B), the OS rate of the IER3 genetic alteration group was not much different from that of the non-genetic alteration group; however, this result was not statistically significant. Figure 6C displayed the relationship between IER3 alterations and DFS in HCC patients. The DFS rate of the IER3 genetic alteration group was higher than that of the non-genetic alteration group; however, this result also failed to reach statistical significance.

Figure 6 Genetic alteration landscapes in HCC tissues.

(A) OncoPrint of IER3 alterations in HCC. (B) The overall survival of IER3 genetic alteration group and non-genetic alteration group. (C) The disease-free survival of IER3 mutant group and non-mutation group. HCC, hepatocellular carcinoma; IER3, immediate early response 3.

Enrichment analysis of DEGs and CEGs upregulated by IER3

Initially, 4,080 upregulated CEGs and 396 upregulated DEGs were identified. After the intersection, 230 genes were positively correlated with IER3 expression (all genes appeared in ≥3 data sets), as shown in Fig. 7A. Figures 7B–7E displays the ranking of the 10 most crucial bubble charts for each project. The biological processes mainly concentrated on cell division, DNA replication, G1/S transition of the mitotic cell cycle, drug response, cell adhesion, mitotic nuclear division, negative regulation of apoptosis, cell proliferation, and cell–cell adhesion. The cellular components were mainly enriched in cytoplasm, nucleus, cytosol, nucleoplasm, extracellular exosome, membrane, extracellular space, nucleolus, extracellular matrix, and the perinuclear region of the cytoplasm. With respect to molecular function, the overlapping genes mainly enriched in protein binding, ATP binding, poly(A) RNA binding, protein kinase binding, identical protein binding, protein kinase activity, cadherin binding involved in cell–cell adhesion, enzyme binding, and protein serine/threonine kinase activity. With respect to the KEGG pathway, the overlapping genes mainly concentrated on cell cycle, the PI3K-Akt signaling pathway, cancer-related signaling pathways, small-cell lung cancer, the p53 signaling pathway, focal adhesion, ECM–receptor interactions, phagosome, oocyte meiosis, and DNA replication. The GSEA analyzed that the GO analyze (Fig. 8) mainly concentrated on mitotic sister chromatid segregation, cell cycle checkpoint signaling, microtubule cytoskeleton organization, DNA replication, and so on. The GSEA analyzed that the KEGG pathways (Fig. 9) mainly concentrated on retinol metabolism, cell cycle, metabolic pathways, and so on.

Figure 7 Enrichment analysis of the upregulated crossover genes of IER3 in HCC.

(A) Intersection Venn diagram of IER3 upregulated CEGs and DEGs. (B) Biological process. (C) Cell component. (D) Molecular function. (E) KEGG pathways. HCC, hepatocellular carcinoma; IER3, immediate early response 3; CEGs, differential expression genes; DEGs, co-expression genes; KEGG, Kyoto Encyclopedia of Genes and Genomes.

Figure 8 GO analyze based on GSEA analyze in HCC.

(A) The overall view in GO analyze; (B) mitotic sister chromatid segregation; (C) cell cycle checkpoint signaling; (D) microtubule cytoskeleton organization. HCC, hepatocellular carcinoma; GSEA, Gene Set Enrichment Analysis; GO, Gene Ontology.

Figure 9 KEGG analyze based on GSEA analyze in HCC.

(A) The overall view in KEGG analyze; (B) retinol metabolism; (C) cell cycle; (D) metabolic pathways. HCC, hepatocellular carcinoma; GSEA, Gene Set Enrichment Analysis; KEGG, Kyoto Encyclopedia of Genes and Genomes.

RFX5 activates the expression of IER3 by binding to the promoter region

Because the upregulation of IER3 was found to be related to shorter survival in HCC, it is necessary to explore the underlying molecular mechanisms of IER3. Many studies have shown that mRNA can be activated by its upstream TFs. A total of 521 TFs that may be related to IER3 were obtained from the hTFtarget database. These putative TFs were crossed with 230 positive genes related to IER3, and we obtained four TFs [regulatory factor X5 (RFX5), lamin B1 (LMNB1), forkhead box M1 (FOXM1), ETV4] that were most closely related to IER3 (Fig. 10A). Ultimately, RFX5 was selected as the possible TF of IER3. In order to verify that IER3 was activated by RFX5 in HCC, the binding information of 2 kb upstream of the IER3 promoter region and TF RFX5 motif were predicted (Fig. 10B). The three motifs with the strongest correlations were TTCCCCAGGAACATC, CTTCCCAGCACCAGA, and CGCCCCAGTCACCGC (Fig. 10C). The results indicated that IER3 had binding peaks of RFX5 in the initial region of translation (Fig. 10D). Figure 11 showed that, compared to non-HCC tissues, RFX5 mRNA in HCC tissues was upregulated (SMD = 0.95, 95% CI [0.46–1.43], p <0.001).

Figure 10 RFX5 activates IER3 transcription by binding to the IER3 promoter region.

(A) Intersection Venn diagram of IER3-associated upregulated genes and related TFs. (B) and (C) The binding site of RFX5 in the promoter region of IER3 and the first five binding sequences. (D) IER3 had binding peaks of RFX5 in the initial region of translation. IER3, immediate early response 3; TF, transcription factor. RFX5, Regulatory factor X5.

Figure 11 Comprehensive expression of RFX5 in HCC calculated with SMD.

Compared to non-HCC tissues, RFX5 mRNA in HCC tissues was upregulated. HCC, hepatocellular carcinoma; SMD, standardized mean difference.

Discussion

HCC remains one of the most malignant tumors worldwide, and effective therapeutic regimens for advanced HCC patients are scarce (Ioannou, 2021; Pang et al., 2020a). Thus, there is an urgent need to classify the mechanisms of HCC formation and deterioration, which would lay the foundation for the development of novel therapeutic agents. Recently, IER3 was shown to be correlated with carcinogenesis. However, few studies have reported the roles of IER3 in HCC. Herein, we confirmed that IER3 is upregulated in HCC and emphasized its potential clinical values. More importantly, this study revealed the prospective molecular mechanisms of IER3 underlying HCC onset and progression.

Compared to other studies on IER3 and HCC, the benefits of our study are as follows. Firstly, our study explored the protein level of IER3 in HCC tissues by using IHC. Additionally, this study included a large number of HCC cases to validate our results; these cases included 1,483 cases of HCC tissues and 1,263 cases of adjacent normal liver tissues. By enlarging the sample sizes, we managed to eliminate the known inter-study batch effects. Lastly, we revealed the potential molecular functions and upstream transcriptional regulation mechanisms of IER3 underlying HCC.

IER3 expression can be induced by various stimuli, and it plays a dual role in cellular effects and a positive or negative role in cell survival. These roles depend on the type of cell, the stimulation received, and the intensity of the stimulation received (Wu et al., 2013). At present, some research teams have studied the role of IER3 in tumors, including liver cancer, bladder cancer (Ye et al., 2018), and pancreatic cancer (Jin et al., 2016). To date, a few related studies have shown that IER3 can be used as a biomarker for cancer prognosis. Furthermore, IER3 was found to play an important role in the regulation of lymph angiogenesis and in the prognosis of tongue cancer; it may, therefore, be a therapeutic target (Xiao et al., 2019). Another study confirmed that IER3 can be used as an independent prognostic factor for bladder cancer. High IER3 expression was significantly correlated with high pathological lymph node staging, and the life expectancy of patients with bladder cancer with overexpressed IER3 was shorter (Ye et al., 2018). The expression of IER3 was significantly correlated with advanced stage and pathological grade in pancreatic cancer (Garcia et al., 2014). In a study of the female reproductive system, IER3 expression and the cancer stage were independent predictors of OS in patients with ovarian cancer (Han et al., 2011). Furthermore, IER3 induced apoptosis in cervical cancer cells, and its expression was downregulated in patients with cervical cancer (Jin et al., 2016). IER3 was a presumed tumor suppressor in the cervix, and the c-Ab1/p73beta/IER3 axis is a new key signaling pathway that endows etoposide with chemosensitivity (Jin et al., 2015).

In HCC, the clinical significance and mechanisms of upregulated IER3 expression in HCC have been elucidated. One study (Liu et al., 2015) detected the expression of IER3 in 62 cases of HCC tumors by IHC and found that IER3 expression was significantly correlated with the expression of P53 and Ki-67, and the maximum diameter and degree of differentiation of the tumor were related to the upregulated expression of IER3. This result was similar to our results, which also indicated that high IER3 expression was correlated to various clinicopathological parameters, including pathological grade. More importantly, the high expression of IER3 indicated poor HCC prognosis and suggests that the upregulation of IER3 may promote the development of HCC.

But our findings must be compared with those of a previous study by Liu and colleagues (Liu et al., 2021), who studied 90 samples of tumors and adjacent normal tissues from patients with HCC. IHC showed that IER3 expression was rarely detected—or detected only at low levels—in HCC relative to normal tissues, and patients with high IER3 expression had a longer OS than patients with low IER3 expression. In contrast, our study demonstrates that higher IER3 expression levels in HCC tissues were associated with poor OS. Although we cannot currently explain this discrepancy in results, we posit that it may be caused by testing IER3 expression at different stages and types of HCC. However, our work clearly found that IER3 was upregulated in HCC tissues, and this upregulation indicated poor OS in HCC patients. Therefore, IER3 may be used as a prognostic marker of HCC.

Although the role of IER3 in some cancers has been reported, the role of IER3 in HCC requires further exploration. In a study of the mechanism of IER3 in HCC, the knockdown of IER3 inhibited the viability, growth, and migration of HCC cells, and the silencing of nuclear protein 1 (NUPR1) could lead to the downregulation of IER3(Emma et al., 2016). ETV4 could regulate cell survival and proliferation through IER3 for HCC cells treated by sorafenib or cisplatin (Chen et al., 2019). IER3 may, nevertheless, be used as a target for promoting the chemosensitivity of tumor therapy. Indeed, the overexpression of IER3 in HeLa cells enhanced their sensitivity to chemotherapy-induced apoptosis (Arlt et al., 2001).

Our study also predicted the TFs that may regulate IER3, and we found that RFX5 had binding domains for the IER3 promoter region. Through KEGG analysis, IER3-related genes were mostly enriched in the PI3K-Akt signaling pathway, cancer-related signaling pathways, the p53 signaling pathway, and other signaling pathways. The p53 signaling pathway is an important pathway in various types of cancers, because it is known to play a regulatory role in a variety of biological functions, such as cell cycle progression, proliferation, migration, invasion, and apoptosis (Duffy et al., 2020). In China, 58% of HBV carriers with HCC also carried p53 mutations (Gao et al., 2019). RFX5 acted as a TF in HCC to regulate the progress of HCC through the p53 signaling pathway. For example, RFX5 promoted the progression of the HCC cell cycle from G0/G1 to the S phase by transcriptional activation of lysine demethylase 4A (KDM4A) and prevented apoptosis in HCC by regulating p53 and its downstream gene targets (Chen et al., 2020). Another study implied that RFX5 inhibited the apoptosis of HCC cells by transactivating tyrosine 3-monooxygenase/tryptophan 5-monooxygenase activation protein theta (YWHAQ). When RFX5 was inhabited, it could significantly downregulate the protein levels of YWHAQ, and the expression of p53 was upregulated in HCC cells following RFX5 silencing (Chen et al., 2019). Therefore, we speculate that RFX5 may bind the IER3 promoter domains and regulate the p53 signaling pathway to enhance the development of HCC. However, the additional experiments needed to determine whether p53 plays a pro-tumorigenic role by recruiting TFs that interfere with certain signaling pathways.

Our current findings not only expanded the knowledge base in this field, but also provided a basis for further research. Some limitations of this study should be considered. Firstly, due to the lack of enough HCC cohort studies with IER3 protein expression levels, the prognostic value of IER3 protein in in-house IHC is not significant. More clinical follow-up and investigations on HCC patients are encouraged to better assess its prognosis. Secondly, the precise molecular mechanisms of RFX5 and IER3 in HCC has not been certified by experiments. In the future, our conclusions need to be validated using both in vitro and in vivo clinical and molecular biological methods. To sum up, our research demonstrated that the expression of IER3 was significantly upregulated in HCC, and that patients with higher IER3 expression level had shorter OS. The upregulation of IER3 may play a crucial role in HCC by recruiting RFX5.

Conclusions

The available results found that higher IER3 expression in HCC patients were associated with poor prognosis in HCC, and that IER3 is be a potential prognostic marker for HCC. The molecular mechanisms of the involvement of IER3 in HCC warrant further study.

Supplemental Information

Supplemental Information 1 Raw data of IER3 protein levels in HCC tissues assessed by in-house IHC

The expression of the IER3 protein was detected by in-house IHC in 94 cases of HCC tissues and in 127 cases of non-HCC liver tissues. The process and interpretation criteria of IHC were listed in the text material methods. IER3: Immediate early response 3; IHC: Immunohistochemistry; HCC: Hepatocellular carcinoma; AFP: Alpha fetoprotein.

Click here for additional data file.

The authors thank the Gene Expression Omnibus (GEO) database, the Oncomine database, The Cancer Genome Atlas (TCGA), the Genotype-Tissue Expression (GTEx) Project, the Database for Annotation, Visualization and Integrated Discovery (DAVID), The Human Protein Atlas, GEPIA website, the Kaplan–Meier Plotter, SurvExpress website, the Biomedical Informatics Institute, cBioPortal for Cancer Genomics, Sangerbox tool, SPSS v.25.0, STATA v.14.0, R v.3.6.1, and GraphPad v.8.0.2.

Additional Information and Declarations

Competing Interests

Author Contributions

Human Ethics

Data Availability

Gang Chen is an Academic Editor for PeerJ.

Fei-Yan He performed the experiments, analyzed the data, prepared figures and/or tables, authored or reviewed drafts of the paper, and approved the final draft.

Gang Chen and Yi-wu Dang conceived and designed the experiments, performed the experiments, analyzed the data, prepared figures and/or tables, authored or reviewed drafts of the paper, and approved the final draft.

Rong-quan He, Zhi-Guang Huang, Wei-Zi Wu and Ji-Tian Chen conceived and designed the experiments, authored or reviewed drafts of the paper, and approved the final draft.

Jian-Di Li analyzed the data, authored or reviewed drafts of the paper, and approved the final draft.

Yu-Lu Tang and Dong-Ming Li analyzed the data, authored or reviewed drafts of the paper, and approved the final draft.

Shang-Ling Pan and Zhen-Bo Feng conceived and designed the experiments, authored or reviewed drafts of the paper, and approved the final draft.

The following information was supplied relating to ethical approvals (i.e., approving body and any reference numbers):

The First Affiliated Hospital of Guangxi Medical University approved this study (2018 KY-E-102).

The following information was supplied regarding data availability:

Data are available at NCBI GEO: GSE12941, GSE84005, GSE45050, GSE64041, GSE117361, GSE54236, GSE63863, GSE65485, GSE73708, GSE81550, GSE87592, GSE174608, GSE114564, GSE148355, GSE69164, GSE104310, GSE63018, GSE77509, GSE94660, GSE97214, GSE112221, GSE101728, GSE98269, GSE10143, GSE14520, GSE33294, GSE46408, GSE46444, GSE57555, GSE60502, GSE125469, GSE128274, GSE166163; TCGA - Liver; GTEx - LIHC.

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
