# Peer review of "Expression of IER3 in hepatocellular carcinoma: clinicopathology, prognosis, and potential regulatory pathways"

_PeerJ, doi:10.7717/peerj.12944_

## Round 0.1 · original submission · Major Revisions

Please provide a comprehensively revised version addressing the editorial comments and a detailed rebuttal letter.

·

Basic reporting

The article presents clear, fluid, and technically correct English, with only a few incorrect verb forms that are necessary to be corrected by authors (p.e., -- in the abstract section, line 36 “were used to explored the molecular mechanisms of IER3” explored instead of explore.)
The article includes sufficient references to contextualize the introduction section and to substantiate the discussion of the work.
The Figures and tables are relevant to the work. However, some of there have enough resolution and/or incorrect information necessary to be corrected by authors:
-- In Figure 1 legend (Figure 1 IER3 protein expression levels in HCC) should be replanted this legend, in this figure is not presented an “expression level” analysis is shown the expression of IER3 in tissue samples of normal liver and HCC by IHC, the analysis of the data of the intensity of the expression is not presented in this caption.
-- In Figure 5, it is necessary to improve the image quality to a major dpi.
-- The 242 line (Fig.3B) must be corrected in the figure caption B and C are in an incorrect order.
The raw data were available.

Experimental design

It is an original investigation under the aims and scope of the journal.
It was Performed with an actual and novel experimental design that interrelates in-silica analysis of gene databases with data obtained from a cohort of patients that allowed obtaining good approximations and answering the questions posed in the design.

Validity of the findings

The approximations obtained from the data analysis are good, the approximations obtained from the data analysis are reasonable, it allows establishing a possible relationship between the RFX5 factor and the IER3 in the context of the HCC.

In the conclusion section line 393 to 395 “RFX5 may bind the IER3 promoter domains and regulate the p53 signaling pathway to promote the incidence and development of HCC” there is not enough experimental data in this work to claim this, is necessary to replant this idea002E

Additional comments

Once the authors have addressed the observations and recommendations described here, I recommend it for publication.

Reviewer 2 ·

Basic reporting

(1) Please check the whole stupid mistakes, such as “Figure 3 (B) sensitivity analysis. (C) Egger’s test.”;
“Figure 1, Fig. 2A.”
(2) The language should be modified. Commercial service or the help of a native English speaker is recommended.
(3) It's a little bit distorted for some figures, such as Figure 3B, 3C. Some of the images have different font sizes, e.g., Figure 5, 6.

Experimental design

(1) For the prognostic analysis, only Kaplan–Meier was performed. Uni-/Multi-factor Cox analyses adjusting other factor are needed for the survival analysis using TCGA, GEO.
Apart from the OS, other types of prognostic status, such as DSS, PFI, should be considered. Only this way, on objective prognostic conclusions can be drawn.
(2) The DEGs and CEGs were obtained from the TCGA? The DEGs from the different datasets of TCGA and GSE may give more reliable information. Additionally, the GSEA should conducted.
(3) Just as stated in discussion, the correlation between IER3 and RFX5 should be analyzed in cellular experiments. The expression pattern of the two proteins can be analyzed in different datasets.
(4) The IHC data of HPA can be considered.
(5) The expression feature of IER3 in different datasets should be analyzed and pooled as well.

Validity of the findings

(1) The detailed P value was not provided in some meta-analysis (such as Figure 3A), Egger’s test (Figure 3B).
(2) After simple check, I did not observe the relatively high expression level in the HCC tissues, compared with paired paracancer tissue. Also, the IHC data of paired cancer and paracancer was not shown. Only two HCC and control data were provided. Since 127 samples were enrolled, the statistical analysis should be performed.

Additional comments

The authors explored the potential correlation between IER3 expression and the clinicopathology, prognosis, and potential regulatory of HCC, based on the collected 94 paraffin masses of HCC and adjacent tissues and 33 cases of normal liver tissues, and the TCGA, GEO public available datasets. The above issues should be concerned.

---

## Round 0.2 · Minor Revisions

Please provide a comprehensively revised version addressing the reviewer's comments and a detailed rebuttal letter

Reviewer 2 ·

Basic reporting

Whether the statement of“IER3’s ??? potential molecular mechanism" in 306 is proper?
"expression levels in HCC tissues was ??? associated with poor OS." in line 367
"When RFX5 is??? silenced, it could significantly downregulate the protein levels of" in 393
”high IER3 expression was related ??? poor prognosis in HCC patients.“ in line 407
Please check the whole text.

Experimental design

"Figure 4 is Uni-/Multi-factor Cox analyses" in the rebuttal letter.
I did not find the cox information in Figure 4.

"the correlation between IER3 and RFX5 should be analyzed in cellular experiments."
The question is not answered directly.

Validity of the findings

no comment

Additional comments

no comment

---

## Round 0.3 · accepted · Accept

Thanks for addressing the minor revisions requested. Now your manuscript is accepted in PeerJ.